# Comparative Exergy and Environmental Assessment of the Residual Biomass Gasification Routes for Hydrogen and Ammonia Production

**DOI:** 10.3390/e25071098

**Published:** 2023-07-22

**Authors:** Gabriel Gomes Vargas, Daniel Alexander Flórez-Orrego, Silvio de Oliveira Junior

**Affiliations:** 1Polytechnic School, University of São Paulo, Av. Luciano Gualberto 380, São Paulo 05508-010, Brazil; soj@usp.br; 2Industrial Process and Energy Systems Engineering, École Polytechnique Fédérale de Lausanne EPFL, Sion, 1950 Valais, Switzerland; 3Faculty of Mines, National University of Colombia, Av. 80 #65-223, Medellin 050034, Colombia

**Keywords:** biomass gasification, decarbonization, bioproducts, exergy analysis, energy integration

## Abstract

The need to reduce the dependency of chemicals on fossil fuels has recently motivated the adoption of renewable energies in those sectors. In addition, due to a growing population, the treatment and disposition of residual biomass from agricultural processes, such as sugar cane and orange bagasse, or even from human waste, such as sewage sludge, will be a challenge for the next generation. These residual biomasses can be an attractive alternative for the production of environmentally friendly fuels and make the economy more circular and efficient. However, these raw materials have been hitherto widely used as fuel for boilers or disposed of in sanitary landfills, losing their capacity to generate other by-products in addition to contributing to the emissions of gases that promote global warming. For this reason, this work analyzes and optimizes the biomass-based routes of biochemical production (namely, hydrogen and ammonia) using the gasification of residual biomasses. Moreover, the capture of biogenic CO_2_ aims to reduce the environmental burden, leading to negative emissions in the overall energy system. In this context, the chemical plants were designed, modeled, and simulated using Aspen plus™ software. The energy integration and optimization were performed using the OSMOSE Lua Platform. The exergy destruction, exergy efficiency, and general balance of the CO_2_ emissions were evaluated. As a result, the irreversibility generated by the gasification unit has a relevant influence on the exergy efficiency of the entire plant. On the other hand, an overall negative emission balance of −5.95 kg_CO2_/kg_H2_ in the hydrogen production route and −1.615 kg_CO2_/kg_NH3_ in the ammonia production route can be achieved, thus removing from the atmosphere 0.901 t_CO2_/t_biomass_ and 1.096 t_CO2_/t_biomass_, respectively.

## 1. Introduction

Biomass is an important source of renewable energy that may help reduce fossil fuel dependence and CO_2_ emissions in the chemical sector. This is especially applicable in the case of Brazil, considering its substantial biomass potential. In recent years, biofuels have accounted for almost 70% of global renewable energy production [1], and biomass was responsible for 25.5% of Brazilian domestic energy supply [2]. This contribution could be boosted further if biomass wastes were converted into valuable energy products such as hydrogen and ammonia. In this way, fossil energy consumption and greenhouse gas emissions could be reduced, while the costs and environmental impact of waste disposal could be relieved. In Brazil, for example, sugarcane and orange bagasses are the primary residues of the sugarcane and juice industries, respectively, which are, in turn, the primary suppliers of bioenergy and juice in the country [2,3]. Typically, bagasse provides combined heat and power production for sugarcane mills. Even though they are well-established procedures in the industry, they are still reasonably inefficient and could be replaced by improved energy conversion processes [4,5]. On the other hand, sewage sludge is a by-product of wastewater treatment that contains various organic and inorganic materials and needs to be disposed of appropriately. In recent years, conventional sewage sludge disposal methods, including landfills and anaerobic digestion, have been adopted. However, these methods take a long time to digest and require large amounts of land. Moreover, they tend to cause environmental issues such as undesirable emissions (e.g., odor and leachate) and the accumulation of heavy metals in soils [6,7].

Biomass gasification has a large potential to simultaneously deal with the treatment of cumbersome wastes while increasing the effectiveness of the utilization of the chemical processes’ byproducts. Several studies have explored the potential of different biomass feedstocks for power generation and biofuel production. Promising results were reported for corn cobs, with a gas production yield of about 2 m^3^/kg, a heating value of 5.6 to 5.8 MJ/m^3^, and a cold gas efficiency between 66% and 68% [8]. Other studies focused on converting olive tree pruning and olive pits into electricity and heat, achieving satisfactory cold gas efficiency (70.7–75.5%) and a favorable calorific value (4.8 to 5.4 MJ kg^−1^) [9]. Vine pruning showed promising results in a 350 kW downdraft gasifier, with a syngas heating value of 5.7 MJ/m^3^, cold gas efficiency of 65%, and power efficiency of 21% [10]. Biohydrogen production from the gasification of agricultural waste through dark fermentation is reportedly an environmentally friendly solution [11]. Hydrogen production from coconut coir and palm kernel shell through air gasification showed substantial hydrogen gas production potential [12]. In this regard, the gasification of agricultural residual biomasses is recognized as a promising method for achieving a sustainable bioeconomy and reducing dependence on fossil fuels, averaging 67% efficiency in energy conversion [13].

Over the last five decades, extensive research has been carried out on biomass gasification, mainly focusing on syngas production [14]. Comprehensive research is underway for the development of cost-effective and energy-efficient gasifiers. Gasifiers can be broadly categorized based on [15]:Fluid dynamics (updraft, downdraft),Modes of heat transfer to the gasification process (auto thermal or directly heated gasifiers and allothermal or indirectly heated gasifiers),Gasification agents (air, oxygen, or steam blown), andPressure (atmospheric or pressurized).

In the above categorization of gasifiers, the classification based on fluid dynamics and modes of heat transfer is of prime importance. Fluid dynamics primarily determines the characteristics of the gases/solids in contact during the gasification process and plays a vital role in influencing the performance of a gasifier. In addition to fluid dynamics, the modes of heat transfer in a gasifier are also important aspects of the study. In the case of a directly heated gasifier, the entire gasification process occurs in a single reactor, and heat evolved from the exothermic reactions is used to carry out endothermic gasification reactions. These gasifiers exhibit several configurations, e.g., fixed bed, fluidized bed, or circulating fluidized bed (operated at temperatures below 900 °C) and entrained flow gasifiers (operated at a higher temperature range of 1200–1500 °C) [16]. The heating values of the product gas from these gasifiers using air and oxygen as gasification agents are in the range of 4–7 MJ/Nm^3^ and 10–12 MJ/Nm^3^, respectively [17].

In contrast, an indirectly heated gasifier consists of two reactors. The heat required for the gasification process is produced by a separate combustion reactor and transported to the reduction reactor using heat-carrying material, such as sand. Syngas obtained from this type of gasifier is rich in CO and H_2_, as the flue gas that is released from the combustion reactor flows separately from the product gas, thus preventing its dilution. This fact results in a higher heating value for the gas (12–20 MJ/Nm^3^) compared to an indirectly heated gasifier. Also, since no oxygen separation unit is necessary and a smaller amount of gas cleaning equipment is installed, a lower capital investment is required [18]. In addition, as the two reactors operate separately, it is easy to control and scale up [19]. The dual fluidized bed (DFB) reactor is a type of indirectly heated gasifier. DFB gasification systems have been studied at laboratory and pilot scales over the last two decades. A DFB gasification plant has been running successfully in Güssing, Austria (8 MWth) since 2001, along with industrial-scale operations in Oberwart, Austria (8.5 MWth) and Ulm, Germany (15 MWth) [20]. Apart from this, the Gothenburg Biomass Gasification (GoBiGas) project in Göteborg, Sweden, has been recently commissioned to produce substitute natural gas (SNG) using wood pellets as feedstock. Finally, the comparative analysis conducted by Florez-Orrego et al. [21] shed light on the emissions profile of fossil fuel and biomass pathways for chemical production. As a result, a negative CO_2_ emissions balance is achieved, indicating a favorable global impact on mitigating atmospheric CO_2_ levels. Notably, the study revealed that for each ton of ammonia produced, approximately 1.7 to 2.3 tons of CO_2_ are effectively sequestered from the environment. Furthermore, the research emphasized the advantageous aspects of utilizing bagasse, despite its indirect emissions. These emissions are offset not only by the captured biogenic emissions but also by the utilization of “greener” electricity imports.

In that regard, gasification is a prominent research topic among the available technological routes in a residual biomass conversion context [22]. The technology could lead to higher energy conversion and production yields [23,24] and reduced sizes for treatment plants [25] and costs [26]. Previous studies have already investigated the use of biomass for synthetic natural gas [27], hydrogen [4], ammonia [21], nitrogen fertilizers [28], and electricity production [29,30]. Some conversion routes are shown in the literature for residual biomass [5,31,32]. However, while different options have already been proposed for each biomass waste, there is a lack of studies dedicated to analyzing the performance of waste upgrade systems and comparing the utilization of all those resources using a common basis defined by thermodynamic and environmental indicators. Thus, this work proposes alternative routes for the conversion of biomass wastes into hydrogen and ammonia, in addition to the optimization and hierarchization of these energy conversion routes. For this purpose, residual biomass with low or no added value will be used, such as sugarcane bagasse, sewage sludge, and orange bagasse. This fact reduces the risk of the perception of biomass utilization as a competitor for food and land resources.

## 2. Process Description

The considered approach integrates a biomass gasification system for agricultural or human wastes, a synthesis gas purification unit, and a conditioning system to produce hydrogen and ammonia. Data was collected through a bibliographic review, in addition to data provided by the Basic Sanitation Company of the State of São Paulo [33]. For the sake of comparison, it was assumed a biomass mass flow rate of 26,400 kg/h. The analysis is focused on determining the minimum energy requirements of those facilities; thus, the composite curves of the chemical systems will be further discussed. Depending on the waste heat available, all heat and electricity requirements should be imported, or part of the energy produced in the form of fuel can be internally consumed. Also, if the amount of waste heat from the exothermic reactions exceeds the domestic demands, power could be internally generated using a waste heat recovery steam network.

### 2.1. Biomass Drying and Chipping Process

In the gasification section, moisture is first removed in a rotary dryer with a specific power consumption of 15 kWh per wet ton of biomass [34]. In this process, the water content of the biomass is reduced to 7% [35]. Furthermore, electricity is used in the chipping process for grinding bagasse to obtain 0.5 mm diameter particles. The power consumption is estimated at 3% of the lower heating value of the biomass input [36]. In order to conduct mass and energy balances for the biomass pre-treatment processes, a FORTRAN subroutine was developed within the Aspen^®^ Plus software [37]. The subroutine was utilized to calculate the quantity of moisture removed in the rotary dryer, denoted as $m_{H_{2}O,removed}$ (kg/h) in Equation (1). This calculation is based on the initial moisture content of the biomass, represented as $\psi_{H_{2}O,moist-bio}$ (%), the desired moisture content of the biomass at the gasifier inlet, denoted as $\psi_{H_{2}O,dry-bio}$ (%), and the mass flow rate of the wet biomass feed, indicated as $m_{H_{2}O,moist-bio}$ (kg/h).
(1)$$m_{H_{2}O,removed}=\left(\psi_{H_{2}O,moist-bio}-\frac{1-\psi_{H_{2}O,moist-bio}}{1-\psi_{H_{2}O,dry-bio}}\times \psi_{H_{2}O,dry-bio}\right)\times m_{H_{2}O,moist-bio}$$

### 2.2. Gasification Process

After the chipping and drying processes, biomass is fed to the gasification unit. A Battelle Columbus Laboratory (BCL) indirect-heated gasifier is adopted (Figure 1) [4,12,18,38]. The gasifier separates the solids from the syngas (i.e., sand and char) and transfers them to a combustion chamber. In this latter case, air is blown to burn the char, which provides heat to the reduction zone. To this end, the hot particles (sand) are separated from the flue gas through a cyclone and recycled back to the reduction bed, ensuring the provision of heat for the endothermic reactions (drying, pyrolysis, and reduction). This approach separates the combustion reactions from the reduction reactions, preventing the dilution of the produced syngas with nitrogen [39]. The temperature in the combustion column reaches approximately 950 °C, while the gasification column operates at a temperature of around 850 °C [4,12,18]. The gasifier operates at atmospheric pressure.

The ultimate and proximate analyses of the dry biomass residues are shown in Table 1. For the calculation of the enthalpy and density of solids, the HCOALGEN and DCOALGEN models are chosen [40]. The gasification process consumes saturated steam as the gasification agent, with a steam-to-biomass ratio of 0.50 [41]. Additionally, combustion air is preheated up to 400 °C [42] to facilitate the combustion of a portion of the char generated. Essentially, in the gasification process, it is crucial to maintain a balance between the heat supplied by the combustion zone and the heat required for the drying, pyrolysis, and reduction stages.

Following the decomposition in the pyrolysis process, the reduction reactions occur in the presence of steam and can be summarized as shown in (R. 1–R. 9) in Table 2.

The syngas produced exits the gasifier and goes through a thermal catalytic cracking process, which converts the produced tar into more desirable compounds [43]. Subsequently, the synthesis gas is cooled down to a temperature of 400 °C. To eliminate impurities that could potentially impact downstream equipment, the gas is subjected to a scrubbing process using water. Following this, the syngas is compressed to 35 bar. To ensure the removal of sulphur compounds, a zinc oxide guard bed is utilized. More information regarding the properties of the mass flows identified in tags 1 to 8 of Figure 1 can be found in the Appendix A.

### 2.3. Syngas Conditioning Process

Upon exiting the gasifier, the syngas undergoes the necessary treatment and adjustment to its composition. This critical process is performed in the syngas treatment unit. In the hydrogen production route, syngas can be directly sent to the water gas shift reactors (Figure 2). However, for ammonia production, it is crucial to achieve an H_2_:N_2_ molar ratio of 3:1. To this end, an autothermal reformer (ATR) is employed, followed by water gas shift reactors, as shown in Figure 3. In the ATR reactor, the partial combustion of the syngas with air enables the introduction of the necessary nitrogen, which provides the energy for the reforming reactions [44]. The reforming reactions consume saturated steam and occur in the presence of a high-temperature-resistant nickel catalyst [45].

The reactions occurring in the ATR involve reactions (R. 2) in Table 2 and (R. 10–R. 11) in Table 3, as well as the combustion of methane, hydrogen, and carbon monoxide. As a result, the molar fraction of the methane slip in the syngas at the ATR outlet is around 0.45% mol [45]. Next, the synthesis gas is cooled to reach an appropriate feed temperature for the downstream high- and low-temperature shift reactors (HT/LT Shift). The recovery of the residual heat is typically achieved by generating high-pressure saturated steam [45].

In the HT shift reactor, an iron-chrome catalyst is used to increase the production of hydrogen by reacting the remaining CO and water in the syngas (R. 11) [46]. The exothermic WGS reaction is limited by equilibrium, which results in a residual CO concentration of approximately 3% mol [45]. To further enhance the conversion of CO, a second WGS reactor is used at a lower temperature (LT Shift) in the presence of a copper–zinc catalyst. The residual CO content at the outlet is typically around 1% [45]. More information about the properties of the mass flows in the syngas conditioning unit can be found in the Appendix A for hydrogen identified in tags 1 to 4 of Figure 2 and Appendix A for ammonia production routes identified in tags 1 to 8 of Figure 3, respectively. Finally, the cooled syngas (35 °C) continues to the syngas purification unit, described in the next section.

### 2.4. Carbon Dioxide Capture and Methanation Processes

A syngas purification unit is needed to remove the carbon compounds produced in the previous sections; otherwise, they could poison the ammonia catalyst. This unit encompasses a CO_2_ capture unit, a methanator, and a dryer, as shown in Figure 4. However, for the hydrogen production route, the methanation unit is spared since no catalyst protection is required, as shown in Figure 5.

In the CO_2_ capture unit of both ammonia and hydrogen production routes, the syngas enters the CO_2_ absorber (35 bar) and is brought into contact with a physical solvent (dimethyl ethers of polyethylene glycols) to form a CO_2_-rich bottom solution. The purified syngas, primarily composed of H_2_ and CO, exits from the top of the absorber column and is sent to the downstream processes. Meanwhile, the pure CO_2_ is gradually released by pressure let-downs through a series of flash drums and expanders, which recover the expansion energy. CO_2_ is sent for transport and disposal at high purity. The lean solvent is recycled back to the absorber [47]. After the CO_2_ removal step, there may still be residual amounts of CO and CO_2_ in the syngas, which need to be eliminated to meet the purity requirements for the ammonia synthesis loop. For this reason, a methanation unit converts those compounds into inert methane by consuming a fraction of the hydrogen over a nickel catalyst. More information about those streams and their properties can be found in the Appendix A for the ammonia production route identified in tags 1 to 4 of Figure 3 and Appendix A for the hydrogen production route identified in tags 1 to 3 of Figure 5.

### 2.5. Pressure Swing Adsorption and Hydrogen Compression in Hydrogen Production Route

The final hydrogen purification is commonly accomplished using pressure swing adsorption (PSA) technology. PSA operates by modulating the internal pressure of adsorption columns, which allows for selective retention of gases (Figure 6). By removing CO_2_ prior to hydrogen purification, the size of the PSA system is reduced, resulting in cost and space savings. The PSA unit operates at 30 bar and 35 °C [4]. The hydrogen recovery efficiency is 95% mol [48]. Pure hydrogen is obtained as the product of the PSA system, while purge gas containing impurities is typically burned to recover the energy. The purified hydrogen is compressed to 200 bar [49]. Appendix A provides detailed information on the streams involved in this unit; see Appendix A identified in tags 1 to 5 of Figure 6.

### 2.6. Ammonia Synthesis Loop

In the integrated route for ammonia production, the final purification stage (Figure 3) results in a syngas with the desired N_2_/H_2_ ratio and a small amount of inert gases. This purified syngas is then compressed at 200 bar and fed to the ammonia synthesis loop (Figure 7). Since reactants are not completely converted in one pass, the unreacted mixture is recompressed and recycled to the ammonia converter. A mixture of fresh and recycled syngas at 200 bar and 35 °C is preheated and introduced into the converter. In the converter, the ammonia synthesis reaction (R. 12) takes place in the presence of an iron-based catalyst, with a fractional conversion that typically ranges between 10% and 30%. The process design and operational parameters are based on refs. [21,45].
(R. 12)$$N_{2}+3H_{2}\;\to 2NH_{3}\;\;\;\;\;({∆h}_{298k}^{0}=92\;\mathrm{kJ}/\mathrm{kmol})$$

The ammonia synthesis is highly exothermic. As a result, moderate temperatures ranging from 350 °C to 550 °C are commonly employed [45] to achieve appropriate equilibrium conversion and an acceptable reaction rate. To effectively manage the temperature and optimize the performance of the ammonia synthesis process, three or more sequential catalytic beds with an intercooling system are adopted. By dividing the reaction into multiple catalytic beds and incorporating intercooling, the temperature can be better managed, reducing the risk of catalyst deactivation and improving overall process performance. Also, this setup enables higher per-pass conversions. After the ammonia synthesis, a significant portion of the produced ammonia is initially condensed using a water-cooling system. However, relying solely on water cooling does not provide satisfactory ammonia condensation. Thus, the unreacted mixture is further cooled to approximately −20 °C to increase the ammonia condensation and the overall efficiency of the ammonia loop. Finally, as an excessive build-up of methane (inert) has negative effects on the reaction conversion and circulation rate, a portion of the hydrogen-rich recycled gas is purged from the system. In this way, the overall inert concentration, including methane, is kept below a suitable threshold, typically 8% mol [45]. The characteristics of the streams associated with this unit are shown in detail in the Appendix A identified in tags 1 to 9 of Figure 7, Appendix A.

### 2.7. Integrated Flowsheets of the Ammonia and Hydrogen Production Routes Using Residual Biomass

Figure 8 and Figure 9 summarize the flowcharts of the hydrogen and ammonia production routes using residual biomass. The distinct features of the two production routes will be responsible for different energy demands, CO_2_ emissions, and chemical production per unit of biomass consumed.

## 3. Materials and Methods

The mass, energy, and exergy balances for each unit operation of the chemical plants are carried out in this work. Indicators based on the exergy concept, namely, the plantwide and extended exergy efficiencies, as well as the CO_2_ balance, are used to assess the hydrogen and ammonia production performance.

### 3.1. Process Modeling

The evaluation of the thermodynamic properties of each process flow, as well as the mass, energy, and exergy balances of each operation unit, is performed using Aspen Plus^®^ V8.8 software [37]. The thermodynamic model used in gasification, treatment, hydrogen production, and ammonia synthesis is Peng-Robinson EOS with Boston–Mathias modifications [50]. On the other hand, for the simulation of the CO_2_ capture unit using DPEG, the thermodynamic model of the theory of the statistical association of the chain (PC-SAFT) is adopted [21,47,51]. The gasification model is composed of sequential pre-treatment (dryer and chipping), pyrolysis, reduction, and combustion processes. Moisture removal is simulated by using a FORTRAN subroutine [21]. The calculation of the mass fractions of volatiles (*x*_j_), condensables, and solids in the pyrolysis reaction step, as well as the gas volume fractions (*v*_i_) of hydrogen, carbon monoxide, carbon dioxide, and methane produced, is carried out using a set of empirical correlations [52]. These correlations are functions of the reaction temperature (T) and are represented by Equations (2)–(8) [52]. A subroutine in MS Excel^®^ integrated into the Aspen simulator performs the atomic balance of species (C, H, O, N, and S, Ash) present in the volatiles, condensables, char, and ash generated during the pyrolysis section.
(2)$$x_{\mathrm{Gas}}=311.10-351.45\left(\frac{T}{500}\right)+121.43{\left(\frac{T}{500}\right)}^{2}\;\;\;\;\;\;\;\;\;\;Gases\;(\%\;mass\;of\;dry\;biomass)\;\;\;\;$$
(3)$$x_{\mathrm{Char}}=-15.03+50.58\left(\frac{T}{500}\right)-18.09{\left(\frac{T}{500}\right)}^{2}\;\;\;\;\;\;\;\;\;\;\;\;\;Char\;(\%\;mass\;of\;dry\;biomass)$$
(4)$$x_{\mathrm{Tar}}=-196.07+300.86\left(\frac{T}{500}\right)-103.34{\left(\frac{T}{500}\right)}^{2}\;\mathrm{Tar}\left(\%\;\mathrm{mass}\;\mathrm{of}\;\mathrm{dry}\;\mathrm{biomass}\right)$$
(5)$$y_{\mathrm{CO}}=240.53-225.12\left(\frac{T}{500}\right)+67.50{\left(\frac{T}{500}\right)}^{2}\;\mathrm{CO}\;(\%\mathrm{mole}\;\mathrm{of}\;\mathrm{gas})\;$$
(6)$$y_{{\mathrm{CO}}_{2}}=-206.86+267.66\left(\frac{T}{500}\right)-77.50{\left(\frac{T}{500}\right)}^{2}\;CO_{2}(\%\mathrm{mole}\;\mathrm{of}\;\mathrm{gas})\;$$
(7)$$y_{{\mathrm{CH}}_{4}}=-168.64+214.47\left(\frac{T}{500}\right)-62.51{\left(\frac{T}{500}\right)}^{2}\;CH_{4}(\%\mathrm{mole}\;\mathrm{of}\;\mathrm{gas})\;$$
(8)$$y_{H_{2}}=234.97-257.01\left(\frac{T}{500}\right)+72.50{\left(\frac{T}{500}\right)}^{2}H_{2}(\%\mathrm{mole}\;\mathrm{of}\;\mathrm{gas})\;$$Compressors and pumps are modeled with 60% and 80% isentropic efficiencies, respectively. The PSA has a hydrogen recovery efficiency of 95% mol [48]. The determination of the chemical exergy adopts the standard environment model proposed by Szargut et al. [53] with reference conditions at T_0_ = 298.15 K and P_0_ = 101.3 kPa. The ratio of specific chemical exergy to lower heating value is calculated using the correlation proposed by ref. [53] for solid fuels with specified mass ratios, Equation (1).
(9)$$\beta =\frac{b^{ch}}{LHV}=\frac{1.0438+0.1882\frac{H}{C}-0.2509\left(1+0.7256\frac{H}{C}\right)}{1-0.3035\frac{O}{C}}$$
whereas the biomass lower heating value (LHV, MJ/kg) is estimated according to Equation (10) [54]:(10)$$LHV=349.1C+1178.3H+100.5S-103.4O-15.1N-21.5ASH-0.0894h_{lv}H$$The mass fractions of carbon (C), hydrogen (H), sulphur (S), oxygen (O), nitrogen (N), and ashes (A) in the dry biomass are reported in Table 1. In addition, h_lv_ is the enthalpy of evaporation of water at standard conditions (2442.3 kJ/kg). The calculated LHV and chemical exergy of biomasses are summarized in Table 4. 

The chemical exergy of a mixture can be calculated using Equation (11) [55]:(11)$${\bar{b}}_{ch,\;mist}=\sum_{i}x_{i}{\bar{b}}_{ch,\;i}+RT_{0}\sum_{i}x_{i}\ln\gamma_{i}x_{i}$$
where ${\bar{b}}_{ch,\;i}$ represents the standard chemical exergy of the substance *i* at $P_{0}$ and $T_{0}$; $x_{i}$ is the molar fraction of the component i; R is the universal gas constant; and $\gamma_{i}$ is the activity coefficient.

### 3.2. CO_2_ Emissions

The general balance of CO_2_ emissions (GBE) is performed according to Equation (3)
(12)$$GBE={{CO}_{2}}_{Direct}^{Biogenic}+{{CO}_{2}}_{indirect}^{Fossil}-{{CO}_{2}}_{Avoided}^{Biogenic}$$
where direct biogenic CO_2_ corresponds to direct emissions derived from the biomass conversion, such as the reactions in the gasifier. Since biomass-derived emissions could be considered circular emissions, the captured biogenic CO_2_ emissions may improve the overall emissions balance by reducing the amount of CO_2_ in the atmosphere (i.e., negative emissions). The indirect fossil CO_2_ emissions consider those emissions that arise from the upstream supply chains of the electricity (62.09 gCO_2_/kWh) [56], the sugarcane bagasse and the orange bagasse (0.0043 gCO_2_/kJ_biomass_) [57], as well as the sludge (0.0106 gCO_2_/kJ_sludge_) [58].

### 3.3. Exergy Efficiency

The overall exergy efficiency of the chemical production routes is evaluated using two performance indicators [4,56], namely the rational and the relative exergy efficiency. The rational efficiency, Equation (13), considers that all the outlets (incl. CO_2_ and purge gas) of the chemical plant are products, while the relative exergy efficiency, Equation (14), is a measure of the deviation from the theoretical exergy consumption when only bio-products are produced in the plant. Thus, the second definition is more conservative, adopting lower values for those processes that produce less useful products.
(13)$$\eta_{rational}=\frac{B_{useful,output}}{B_{input}}=1-\frac{B_{Dest}}{B_{input}}=1-\frac{B_{Dest}}{B_{biomass}+W_{imported}}$$
(14)$$\eta_{relative}=\frac{B_{consumed,ideal}}{B_{consumed,actual}}=\frac{B_{bioproduct}}{B_{biomass}+W_{imported}}$$
where *B* is the exergy flow rate (kW) and *B_Dest_* represents the exergy destruction rate. *W* is the electrical power imported from the grid. The bio-product refers either to hydrogen or ammonia. 

### 3.4. Definition of the Optimization Problem

The minimum energy requirement (MER) is calculated using the OSMOSE Lua platform developed at the IPESE group of the Federal Polytechnique School of Lausanne—EPFL, in Switzerland [59]. To calculate the MER, each hot and cold stream contribution to the overall heat balance is considered and incorporated into the respective hot and cold composite curves. The minimum temperature difference (ΔT_min_) concept is employed to ensure reasonable heat transfer rates, and its value varies depending on the characteristics of the heat flow. For gaseous, liquid, and two-phase flows, a respective temperature difference contribution of 8 °C, 5 °C, and 2 °C is adopted [60]. The objective function and the associated constraints of the MER optimization problem are shown in Equations (15)–(17):(15)$${Min}_{R_{r}}R_{N_{r}+1}$$Subject to heat balance of each interval of temperature *r*
(16)$$\sum_{i=1}^{N}Q_{i,r}+R_{i,r}-R_{r}=0\forall_{r}=1\ldots N$$
(17)$$\mathrm{Feasibility}\;\mathrm{of}\;\mathrm{the}\;\mathrm{solution}\;R_{r}\geq 0$$
where *N* is the number of temperature intervals defined by considering the supply and the target temperatures of the entire set of streams, and *Q* is the heat exchanged between the process streams (*Q_i,r_* > 0 hot streams, <0 cold streams). Finally, *R* is the heat cascaded from higher (*r + 1*) to lower (*r*) temperature intervals (kW).

## 4. Results and Discussion

Due to its impact on global process energy efficiency and chemical yield, the gasification system is considered the most important unit. Thus, the results obtained from the simulation of the gasification system were validated using the study conducted by Marcantonio et al. [41] using walnut husk (M_db_: 12%, Ash_db_: 1.2%, VM_db_: 80.6%, FC_db_: 18.2%, C: 47.9%, H: 6.3%, N: 0.32%, O: 44.27%, S: 0.015%). The comparative results shown in Figure 10 show good agreement with the reported study. The most significant deviation was found for CO_2_ (5%), whereas, for the other substances, the error of the simulation was less than 3%. It can be attributed to the inherent complexities of the gasification reactions and the uncertainties associated with biomass composition.

According to Figure 11, among the investigated biomass residues, orange peel gasification exhibits the highest cold gas efficiency (80.66%), which implies that a substantial portion of the energy content in the waste material is effectively converted into syngas. Conversely, sugarcane bagasse gasification shows the highest carbon conversion efficiency (92.88%), indicating that a major proportion of the carbon in the residual biomass is successfully converted into syngas components. However, it is also important to mention that sugarcane bagasse conversion exhibits the lowest cold gas efficiency (77.62%) among the studied configurations.

Table 5 and Table 6 present a breakdown of the exergy destruction among the main equipment and processes of the different biomass conversion routes. As expected, the gasifier contributes the largest share of exergy destruction in the plants. The biomass grinding and drying and syngas scrubbing and compression are also accounted for as part of the gasification unit, which is in agreement with other studies [21,38].

Considering the different residual biomass conversion routes, the process that presents the highest fraction of exergy destruction in relation to the total exergy destruction in the plant was the orange bagasse gasifier for ammonia conversion (73.6%), as shown in Table 5. On the other hand, gasification via sugarcane bagasse has the lowest exergy destruction share (68.1%). According to Table 5, the ATR only contributes 2.0% to the total exergy destruction in the plant, despite the partial combustion of the produced syngas. Compression systems have relatively high participation in the irreversibility of the whole energy conversion system (5%), especially in the case of ammonia production via sugarcane bagasse. This circumstance is due to the fact that a large amount of syngas compression entails the loss of valuable energy in the form of waste heat. A way to help reduce the amount of exergy destroyed in Thomass-based production plants is to employ better technologies to remove bagasse moisture as well as implement hot catalytic cleaning of the syngas, thus avoiding the waste heat in the water scrubbing section. An increase in the gasifier pressures would also help avoid excessive compression power consumption [61].

The difference in the exergy destruction in the hydrogen (Table 5) and ammonia (Table 6) production routes can be partly explained by the irreversible combustion process of the purge gas in the former route, along with higher power consumption by the hydrogen compression and export system. The gasifier’s relative contribution to the overall exergy destruction is thus smaller in the context of the hydrogen conversion routes. It should also be kept in mind that the amount of exergy recovered per ton of ammonia produced is higher than in the case of hydrogen production routes, even though the latter route has a smaller number of unit operations.

The plantwide exergy efficiency (i.e., without considering the supply chain efficiency), shown in Figure 12, also exhibits this trend. The performance of the hydrogen production route is lower than that of the ammonia production route due to a more stringent purification system and higher compression levels. A large production of offgas and its flaring impairs further its exergy efficiency [29]. Compression and intercooling also require a significant amount of energy input per unit of hydrogen produced. In contrast, when liquid ammonia is expanded, energy can be harnessed through expansion, thus partially recovering the compression power [61,62].

### 4.1. Energy Integration Analysis and Power Generation Potential

The energy integration approach relies on pinch analysis methodology to maximize waste heat recovery throughout the plant. It allows for calculating the minimum energy requirements (MER) of the chemical processes. From the analysis of the composite curves presented in Figure 13, enough waste heat is available from the biomass conversion routes of agricultural waste and sewage sludge, avoiding the need for additional fuel imports. It will still need an additional cooling requirement, such as that provided by a cooling water system.

On the other hand, the electricity requirements, such as compression, refrigeration, pumping, and grinding, could be satisfied either by importing renewable electricity from the electricity mix or by self-generating some power using a Rankine cycle. Waste heat available from the chemical systems also suggests opportunities for providing waste heat to a nearby urban settlement or, depending on the temperature levels, using the waste heat to produce refrigeration using absorption refrigeration systems. The total amount of waste heat cascading available at a high temperature can be better appreciated from Figure 14a–f. In order to quantify the potential power generation using a Rankine cycle, a temperature of waste heat recovery steam generation of 400 °C and a condensation temperature of 25 °C are adopted. Assuming a realistic Carnot efficiency of 50% and considering the waste heat cascade shown in Figure 14a–f, the power generated in a Rankine cycle-based power plant operating at the mentioned temperatures can be calculated and is reported in Table 7. Major differences between the power generation potential of the ammonia and the hydrogen production routes are observed. The potential for power generation in the ammonia production routes is higher than in hydrogen production. 

The exothermic reactions involved in the ammonia synthesis contribute to the higher power output observed in the ammonia production route. Among the biomass residues studied, the orange peel biomass exhibits the highest potential for power output (15,171 kW), while the sugarcane biomass conversion route shows the lowest potential for power output (7259 kW), which can be attributed to the properties of the biomass used. The hydrogen production routes show lower power generation potential. The orange peel conversion has the highest potential for power generation (13,735 kW) when used to produce hydrogen, while the sugarcane conversion route has a small power generation potential (6208 kW) in a Rankine cycle-based power plant when used to produce hydrogen. These results highlight the influence of biomass composition and its conversion process on the r generation potential.

### 4.2. General CO_2_ Emissions Balance

Figure 15a,b, and Table 8 and Table 9 summarize the results of the balance of CO_2_ emissions for each biomass-based chemical production route. As it can be seen, the indirect fossil contributions to the emissions balance are not negligible, which reveals environmental burdens that might otherwise remain hidden if imported electricity or biomass were considered emission-free inputs. The indirect emissions from the sewage sludge supply chain (3.89 kg_CO2_/kg_H2_) are the largest among all the chemical production routes. Nevertheless, all the hydrogen production routes using any residual biomass present an overall negative balance of emissions, as the avoided emissions offset the effect of the indirect ones. Among the hydrogen production routes, the conversion route of orange peel showed the best performance in terms of negative emissions. The biomass conversion plants captured a significant amount of emissions along the supply chain, thus making a positive contribution to the environmental impact. Considering the CO_2_ emissions balances for the chemical processes of ammonia production (Figure 8b), the conversion route using sewage sludge presents the worst performance in terms of emissions balance, although negative emissions can still be obtained (−0.448 kg_CO2_/kg_NH3_). The biomass conversion route of orange peel to produce ammonia proved to be the best solution for the improvement of the global emissions balance (−1.615 kg_CO2_/kg_NH3_). It is worth mentioning that the conversion routes using sugarcane bagasse have shown excellent performance in terms of CO_2_ emissions reduction. The utilization of these biomasses for producing hydrogen and ammonia as value-added products shows negative values for the emission balance for all the conversion routes.

## 5. Conclusions

In this work, the use of residual biomass gasification in integrated chemical production plants is presented. The energy integration and extended exergy analyses allowed us to point out the opportunities to maximize the recovery of available waste heat exergy throughout the plant. As a result, the implementation of a Rankine cycle allowed the recovery of residual heat from biomass conversion in the ammonia production route, resulting in a potential power output of approximately 15,171 kW. Similarly, in the hydrogen production route, the power generation potential reached 13,735 kW. The sugarcane bagasse-based route shows the highest hydrogen yield rate (40.32 t H_2_ per day) and the largest ammonia production rate (237.43 t NH_3_ per day). The exergy efficiencies calculated ranged from 39% to 43% for hydrogen production routes and from 46% to 57% for ammonia production routes. The overall emission balances ranged from −0.226 to −5.953 kg_CO2_/kg_H2_ and −0.448 to −1.615 kg_CO2_/kg_NH3_, respectively. Negative values point towards the environmental benefit of producing chemical products through residual biomass by depleting CO_2_ from the atmosphere. Many efforts in the research and development of technologies for more efficient conversion of renewable energy sources should aim to boost alternative routes of production of chemicals at larger scales. It should be noted that by defining the extended plant consumption and the extended efficiency concepts, the real effect of the production process, including the upstream supply chain inefficiencies, can be assessed. In this way, the results proved to be strongly dependent on the indirect fossil emissions of those supply chains. In fact, the contribution to atmospheric emissions is not negligible, and it reveals environmental issues that might otherwise remain hidden if imported electricity or biomass were considered emission-free energy inputs.

## Figures and Tables

**Figure 1 entropy-25-01098-f001:**
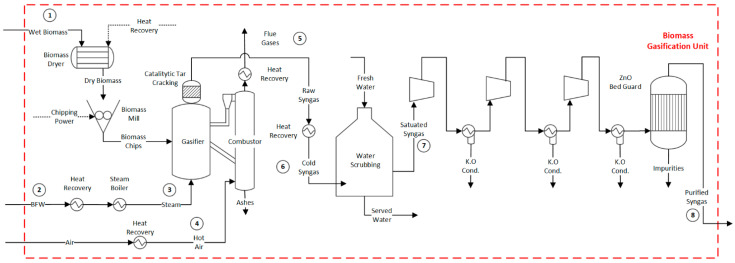
Flowsheet of the biomass pre-treatment and gasification unit. See Appendix A for numbered stream properties. Flow properties (1–8) can be found in the Appendix A.

**Figure 2 entropy-25-01098-f002:**
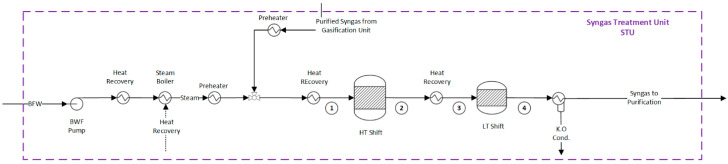
Flowsheet of the syngas conditioning unit for hydrogen production. Flow properties (1–4) can be found in the Appendix A.

**Figure 3 entropy-25-01098-f003:**
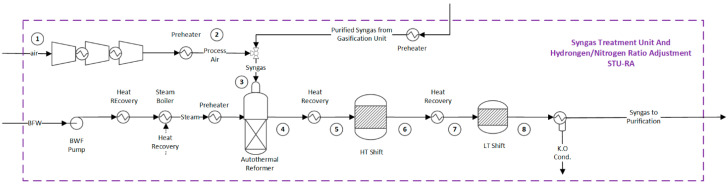
Flowsheet of the syngas treatment unit for ammonia production. Flow properties (1–8) can be found in the Appendix A.

**Figure 4 entropy-25-01098-f004:**
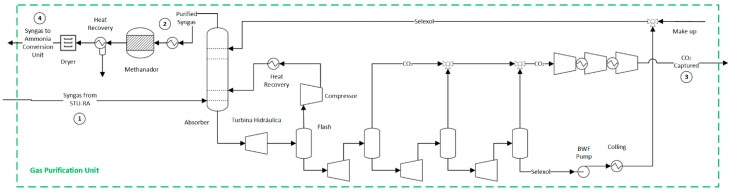
Flowsheet of the syngas purification unit for the ammonia production route. Flow properties (1–4) can be found in the Appendix A.

**Figure 5 entropy-25-01098-f005:**
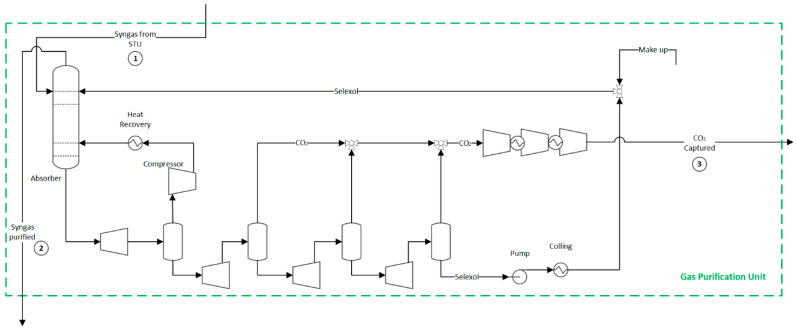
Flowsheet of the syngas purification unit for hydrogen production. Flow properties (1–3) can be found in the Appendix A.

**Figure 6 entropy-25-01098-f006:**
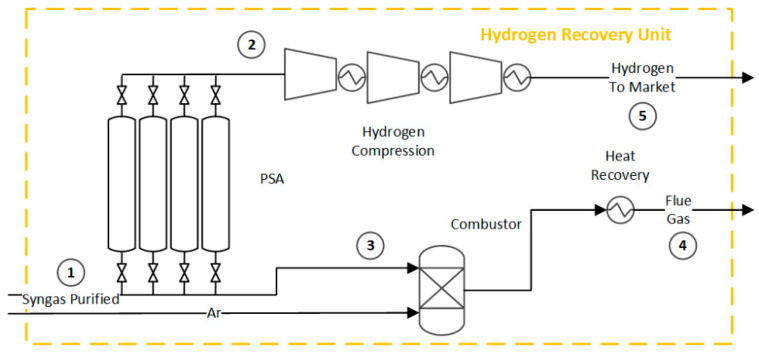
Flowsheet of the pressure swing adsorption for hydrogen recovery and purge gas combustion. Flow properties (1–5) can be found in the Appendix A.

**Figure 7 entropy-25-01098-f007:**
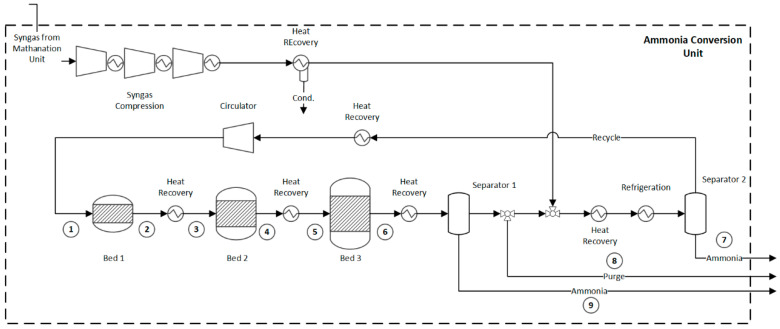
Flowsheet of the ammonia synthesis loop. Flow properties (1–9) can be found in the Appendix A.

**Figure 8 entropy-25-01098-f008:**
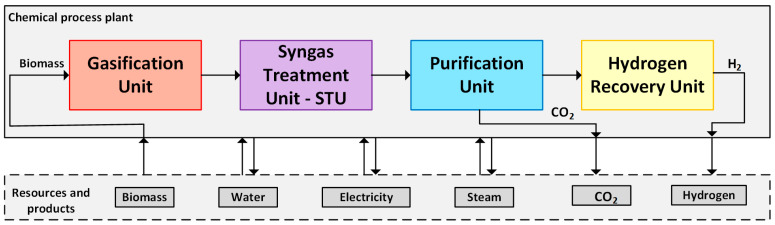
Flowcharts of the hydrogen production route from residual biomass.

**Figure 9 entropy-25-01098-f009:**
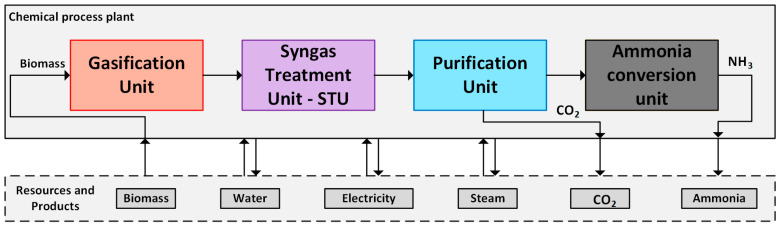
Flowcharts of the ammonia production route from residual biomass.

**Figure 10 entropy-25-01098-f010:**
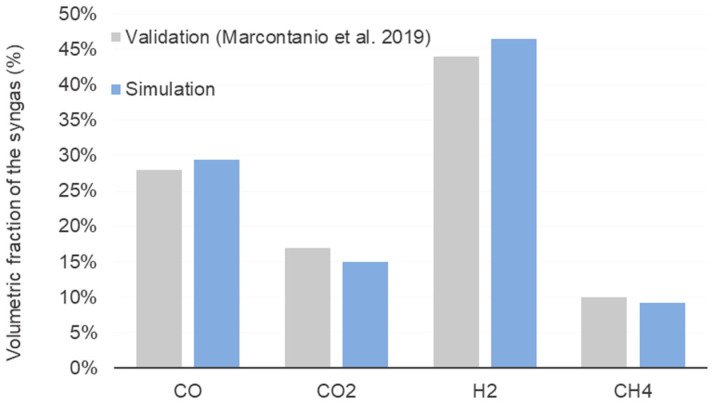
Comparison of the biomass gasification modeling results between the simulation in this work and the literature data reported by Marcantonio et al. [41] for walnut husk.

**Figure 11 entropy-25-01098-f011:**
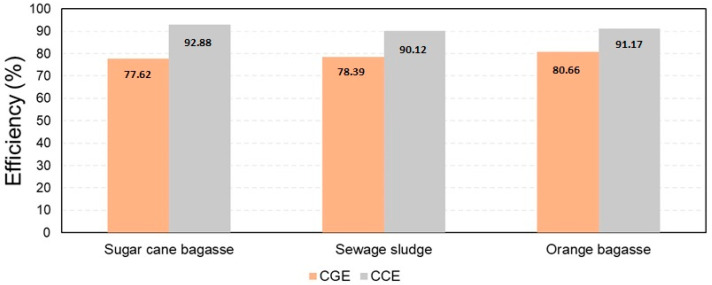
Cold gas efficiency (CGE) and carbon conversion efficiency (CCE) in the gasification process of biomass residues.

**Figure 12 entropy-25-01098-f012:**
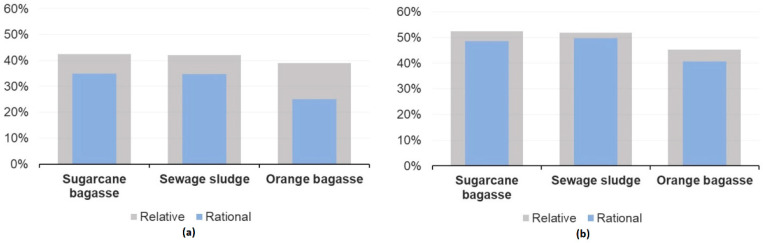
Comparison of the exergy efficiencies of (**a**) hydrogen and (**b**) ammonia production routes using different types of residual biomass.

**Figure 13 entropy-25-01098-f013:**
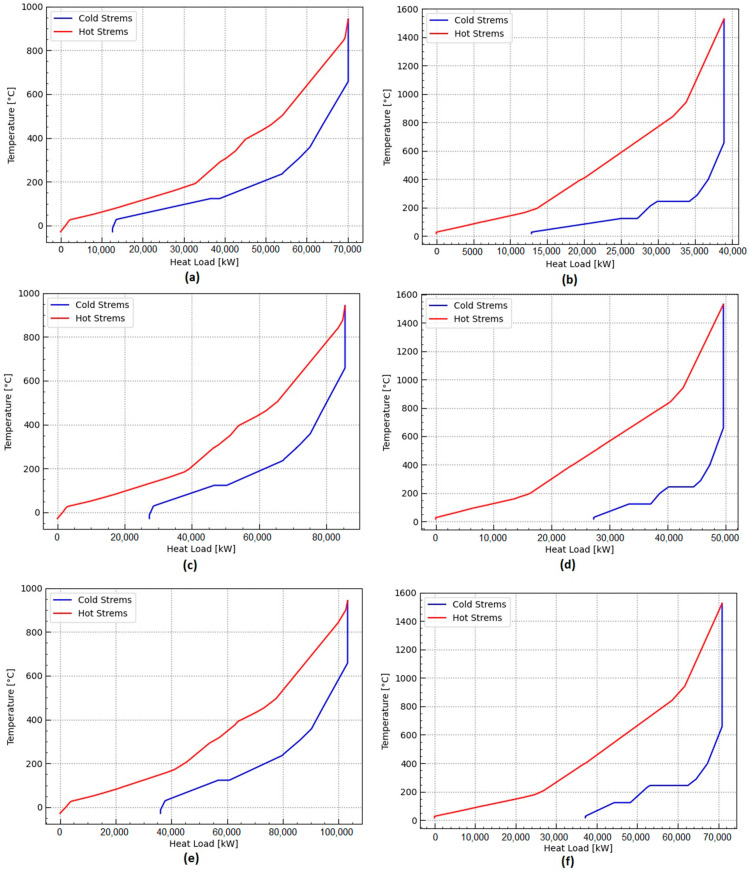
Cold and hot composite curves for different waste biomass conversion processes exhibiting no need for external heating requirements, but showing the need for further cooling requirements: (**a**) sugarcane bagasse to hydrogen; (**b**) sugarcane bagasse to ammonia; (**c**) sewage sludge to hydrogen; (**d**) sewage sludge to ammonia; (**e**) orange bagasse to hydrogen; (**f**) orange bagasse to ammonia.

**Figure 14 entropy-25-01098-f014:**
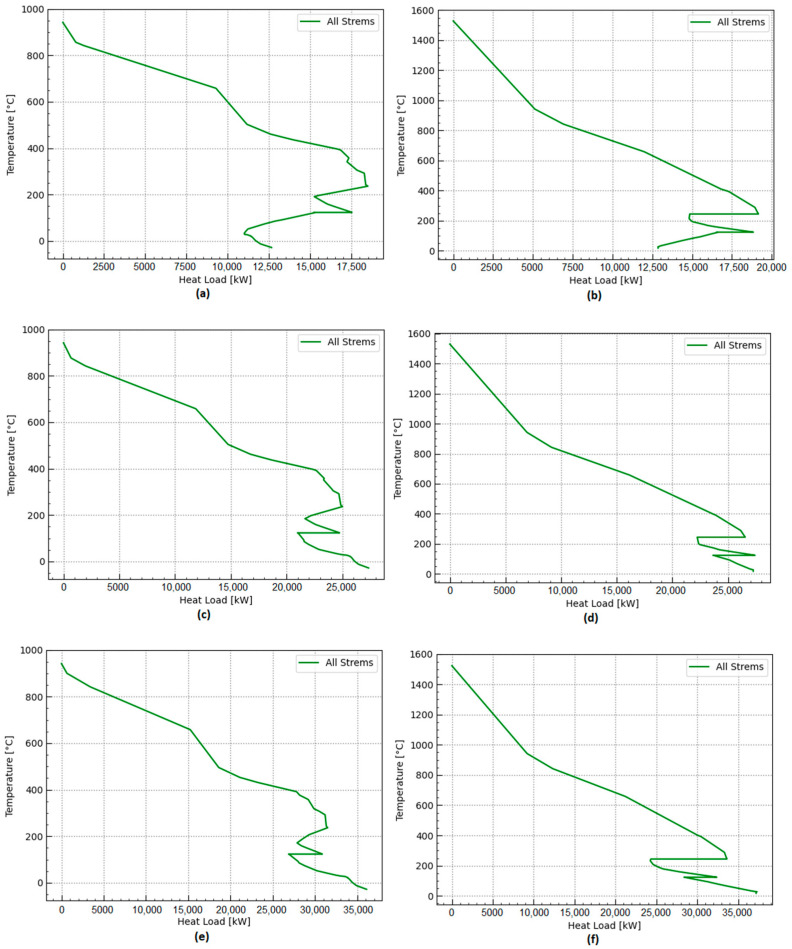
Grand composite curves for different waste biomass conversion processes exhibiting no need for external heating requirements, but showing the need for further cooling requirements: (**a**) sugarcane bagasse to hydrogen; (**b**) sugarcane bagasse to ammonia; (**c**) sewage sludge to hydrogen; (**d**) sewage sludge to ammonia; (**e**) orange bagasse to hydrogen; (**f**) orange bagasse to ammonia.

**Figure 15 entropy-25-01098-f015:**
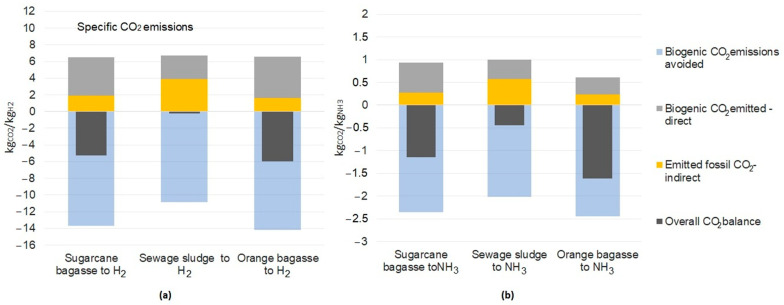
General and detailed emissions (biogenic and fossil, emitted directly, indirectly, and avoided) for the conversion process of (**a**) hydrogen and (**b**) ammonia for different types of selected biomass.

**Table 1 entropy-25-01098-t001:** Proximate and ultimate analyses of different waste biomass used in the gasification process (%). db stand for dry basis.

Parameter	Sugar Cane Bagasse [43]	Sewage Sludge [44]	Orange Bagasse [45]
Proximate analysis			
Fixed Carbon	50.00	18.40	9.23
Volatile Material	14.32	7.60	73.20
Moisture	83.54	64.90	20.60
Ash	2.14	27.50	6.20
Ultimate Analysis (%) ^db^			
Carbon	46.70	33.90	46.40
Hydrogen	6.02	6.30	5.54
Oxygen	44.95	25.50	40.15
Nitrogen	0.17	5.88	1.70
Sulphur	0.02	0.67	0.00
Chlorine	0.00	0.21	0.00

**Table 2 entropy-25-01098-t002:** General reactions of the gasification process.

Reaction	$∆H_{298K}^{0}$ (kJ/mol)	Name	No.
C + O_2_ → CO_2_	−394	Complete combustion	(R. 1)
C + CO_2_ → 2CO	+173	Boudouard reaction	(R. 2)
C + H_2_O → CO + H_2_	+131	Char steam gasification	(R. 3)
C + 2 H_2_ → CH_4_	−75	Char gasification	(R. 4)
CO + ½ O_2_ → CO_2_	−283	Carbon oxidation	(R. 5)
H_2_ + ½ O_2_ → H_2_O	−242	Hydrogen oxidation	(R. 6)
CH_4_ + 2 O_2_ → CO_2_ + 2H_2_O	−283	Methane oxidation	(R. 7)
CO + H_2_O → CO_2_ + H_2_	−41	Water-gas shift reaction	(R. 8)
6CO + 9H_2_ → 6H_2_O + C_6_H_6_	−1583	Tar formation	(R. 9)

**Table 3 entropy-25-01098-t003:** Reforming and water gas shift reactions in the ATR.

Reaction	$∆H_{298K}^{0}$ (kJ/mol)	Name	
CH_4_ + H_2_O → CO + 3H_2_	+206	Steam reform	(R. 10)
CO + H_2_O → CO_2_ + H_2_	−41	Water gas shift reaction	(R. 11)

**Table 4 entropy-25-01098-t004:** Calculated lower heating value (LHV) and specific chemical exergy (b^CH^) for the selected waste streams used in the gasification process.

Biomass	LHV (MJ/kg)	b^CH^ (MJ/kg)
Sugar cane bagasse	17.39	19.50
Sewage sludge	19.25	16.13
Orange bagasse	25.24	20.26

**Table 5 entropy-25-01098-t005:** Breakdown of the exergy destruction in the biomass to ammonia conversion routes.

	Sugar Cane Bagasse	Sewage Sludge	Orange Bagasse
Gasification (%)	68.1	69.8	73.6
Chipping (%)	2.0	2.1	2.2
Dryer (%)	3.5	2.6	1.6
Scrubber (%)	3.6	3.9	2.7
ATR (%)	2.3	2.5	2.1
Shift reactors (%)	0.9	0.9	0.8
Physical absorption (%)	3.4	2.9	3.3
Methanator (%)	0.2	0.2	0.3
Compression (%)	4.3	3.3	2.4
Ammonia reactors (%)	4.0	4.0	3.1
Others (%)	7.7	7.8	7.9

**Table 6 entropy-25-01098-t006:** Breakdown of the exergy destruction in the biomass to hydrogen conversion routes.

	Sugar Cane Bagasse	Sewage Sludge	Orange Bagasse
Gasification (%)	56.7	57.8	63
Chipping (%)	4.6	4.6	4.3
Dryer (%)	2.6	1.9	1.2
Scrubber (%)	2.7	2.9	2.1
Compression (%)	2.4	2.7	1.9
Shift reactors (%)	0.3	0.3	0.2
Physical absorption (%)	7.0	5.9	6.0
PSA combustor (%)	19.9	19.8	17.6
Others (%)	3.8	4.1	3.7

**Table 7 entropy-25-01098-t007:** Power generation potential using a Rankine cycle-based power plant with a waste heat recovery steam generator at 400 °C and a condenser at 25 °C that recovers heat throughout the chemical production plants.

Chemical Plant	Power Generated
Sugarcane bagasse to hydrogen	6208 kW
Sugarcane bagasse to ammonia	7259 kW
Sewage sludge to hydrogen	11,835 kW
Sewage sludge to ammonia	13,147 kW
Orange bagasse to hydrogen	13,735 kW
Orange bagasse to ammonia	15,171 kW

**Table 8 entropy-25-01098-t008:** CO_2_ emissions and other exergy consumption remarks for hydrogen production using different types of waste biomass.

Process Parameter	Sugarcane Bagasse	Sewage Sludge	Orange Bagasse
Biomass Consumption (t_biomass_/t_H2_)	27.39	20.54	15.86
Syngas produced in the gasifier (MJ/kg_H2_)	188.82	187.96	237.24
Hydrogen Produced (t_H2_/day)	23.32	31.13	40.32
Heating requirement ^1^ (GJ/t_H2_)	0.00	0.00	0.00
Cooling requirement ^1^ (GJ/t_H2_)	47.60	75.66	83.64
Captured CO_2_ (t_CO2_/t_biomass_)	0.503	0.534	0.901
Fossil CO_2_ emitted—indirect ^2^ (kg_CO2_/kg_H2_)	1.919	3.896	1.629
Indirect emitted CO_2_—EE (%)	0.081	0.078	0.072
Indirect emitted CO_2_—Biomass (%)	0.919	0.922	0.928
Total fossil CO_2_ emitted (kg_CO2_/kg_H2_)	1.919	3.896	1.629
Biogenic CO_2_ emissions avoided ^3^ (kg_CO2_/kg _H2_)	13.682	10.869	14.166
Biogenic CO_2_ emitted—direct (kg_CO2_/kg _H2_)	6.527	6.747	6.584
Total atmospheric emissions (kg_CO2_/kg _H2_)	8.447	10.643	8.213
General balance of CO_2_ emissions ^4^ (kg_CO2_/kg _H2_)	−5.235	−0.226	−5.953

^1^—Chemical process heating requirements (energy basis) determined from the composite curves. ^2^—Considers indirect emissions due to sewage sludge (0.0106 gCO_2_/kJ_sludge_) [58], electricity (62.09 gCO_2_/kWh), and residual bagasse (0.0043 gCO_2_/kJ_biomass_) supply chains [46,56]; ^3^—CO_2_ emissions captured through the physical absorption system; ^4^—considers the total CO_2_ emitted (fossil or biogenic) minus the biogenic CO_2_ captured.

**Table 9 entropy-25-01098-t009:** CO_2_ emissions and other exergy consumption remarks for ammonia production using different types of waste biomass.

Process Parameter	Sugarcane Bagasse	Sewage Sludge	Orange Bagasse
Biomass Consumption (t_biomass_/tN_H3_)	3.93	3.05	2.26
Syngas produced in the gasifier (MJ/kg_NH3_)	28.02	27.93	34.99
Ammonia produced (t_NH3_/day)	157.16	209.47	273.43
Heating requirement ^1^ (GJ/t_NH3_)	0.00	0.00	0.00
Cooling requirement ^1^ (GJ/t_NH3_)	6.72	11.28	11.36
Captured CO_2_ (t_CO2_/t_biomass_)	0.603	0.668	1.096
Fossil CO_2_ emitted—indirect ^2^ (kg_CO2_/kg_NH3_)	0.272	0.572	0.230
Indirect emitted CO_2_—EE (%)	0.101	0.093	0.091
Indirect emitted CO_2_—Biomass (%)	0.899	0.907	0.909
Total fossil CO_2_ emitted (kg_CO2_/kg_NH3_)	0.272	0.572	0.230
Biogenic CO_2_ emissions avoided ^3^ (kg_CO2_/kg_NH3_)	2.351	2.022	2.450
Biogenic CO_2_ emitted—direct (kg_CO2_/kg_NH3_)	0.936	1.003	0.606
Total atmospheric emissions (kg_CO2_/kg_NH3_)	1.208	1.574	0.835
General balance of CO_2_ emissions ^4^ (kg_CO2_/kg_NH3_)	−1.142	−0.448	−1.615

^1^—Chemical process heating requirements (energy basis) determined from the composite curves. ^2^—Considers indirect emissions due to sewage sludge (0.0106 g_CO2/_kJ_sludge_) [58], electricity (62.09 g_CO2_/kWh), and residual bagasse (0.0043 g_CO2_/kJ_biomass_) supply chains [46,56]; ^3^—CO_2_ emissions captured through the physical absorption system; ^4^—considers the total CO_2_ emitted (fossil or biogenic) minus the biogenic CO_2_ captured.

## Data Availability

Not applicable.

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
