# Peer review of "Comparative Exergy and Environmental Assessment of the Residual Biomass Gasification Routes for Hydrogen and Ammonia Production"

_entropy, 2023, doi:10.3390/e25071098_

Round 1

Reviewer 1 Report

The study provides an exergy assessment of converting different biomass sources to ammonia or hydrogen. My comments to the authors are the following:

Abstract: Absolute values for CO2 emissions lack meaning if there is no idea of the scale of the plants.

Introduction: what gasification technologies are suitable for biomass, what produciton scales are aimed at. Discuss challenges regarding feedstock biomass availability. Compare the negative emissions achieved with biomass relative to fossil fuel routes (natural gas or coal)

Process superstructure: why is there a need for an autothermal reformer? What is the syngas methane fraction or heavies content? What temperature is achieved in the gasification section, what carbon conversions are achievable? Syngas compositions and cold gas efficiencies would be illustrative for the reader to better understand the first conversion step better.

The process models are based on the same mass input of biomass to the plants. What are the implications in terms of energy input, and specific energy consumption. An energy breakdown would be helpful to understand the different forms of energy exchanged in the process. Is there a power cycle to generate electricity from steam produced from syngas cooling and exorthermic synthesis reaction, or is it all used for gasification.

Regarding the exergy methodology, what is assumed as a reference environment (compositions etc.). How is the chemical exergy of gaseous species determined? Why are the reported exergy efficiencies to H2 lower than for NH3, given that the latter requires an extra exothermic conversion step. In Figure 3, is there no exergy desstruction related to heat recovery?

Please give a better explanation as to why the gasifier presents such a large exergy destruction contribution relative to the remaining elements (possibly the cold gas efficiency is very low)

Please check the grammar 

Reviewer 2 Report

·       Please review Lines 217-219. The authors mention that the highest proportion of exergy destruction for gasification has been found for orange bagasse valorisation for ammonia conversion (73.6%). However, the data (73.6%) refers to orange bagasse for hydrogen production (Figure 3a).

·       For the proportion of exergy destruction of the gasification process for the two processes (hydrogen and ammonia production), why is the exergy destruction for gasification in hydrogen production tend to be higher than ammonia production for the same biomass? This can be discussed further to better explain the plantwide performances in Figure 4.

·       Please review lines 238-239. The authors claim that the performance of ammonia production is impaired compared to the hydrogen production route. However, data (Figure 4) shows better exergy efficiencies for ammonia production than hydrogen production.

·       Data in Section 4.1 (Figure 5) can be discussed further. The authors generally discussed the availability of waste heat and possible means of maximizing heat recovery for the process. However, there is a lack of comparison between the results of the two processes. Which of the two processes showed better potential in terms of minimum energy requirements?

·       The conclusions include the important numerical values found in the study for the two processes. However, the concluding remark on the overall comparison of the two production routes, considering the performance indicators measured, is lacking.

Reviewer 3 Report

- Please, include the novelty of the present work compared with the recently paper published in Computers and Chemical Engineering “Multi-time integration approach for combined pulp and ammonia production and seasonal CO2 management”

- Correct the format style of Table 1.

- What is the nature of the gasifier used in the simulation? Fixed bed gasifier? Fluidized bed gasifier? Please explain more in detail about the gasifier reactor type and the assumptions considered in the simulation.

- What is the tar production in this model compared with the tar production in the experimental analyses? Tar content in the syngas is very important for the future syngas upgrading phase. There is no tar description and its importance in the syngas cleaning phase and post treatment. Include the tar modelling equation adopted in the paper.

- To show the exergy breakdown in bar plot is nor very understandable for readers. It is better to use a Sankey diagram

- The methodology content about syngas production, biomass gasification mathematical modelling, hydrogen production and syngas pos-treatment is in general quite poor. Many technical information is missing in the present work: table about the performance parameters of the gasifier, gasification temp, tar modelling, operating pressure and heat losses, etc,

- Quality of Fig.5 should be increased (vectorial image)

-

Round 2

Reviewer 1 Report

The authors have addressed the issues and the article is publishable.

Reviewer 2 Report

The revised manuscript has addressed the reviewers' concerns and comments adequately, thus is recommended for consideration of possible publication. 

Reviewer 3 Report

Comments well executed